# Fast Inference of Visual Autoregressive Model with Adjacency-Adaptive Dynamical Draft Trees

## Abstract

Autoregressive (AR) models have made significant strides in image generation, delivering quality comparable to diffusion-based methods. However, their sequential inference process incurs high computational costs, hindering efficiency and scalability. Although speculative decoding has proven effective in accelerating Large Language Models (LLMs), its adaptation to visual AR models, especially for improved generation with dynamic draft trees, remains largely unexplored. In this work, we identify a key obstacle in applying speculative decoding to visual AR models: inconsistent acceptance rates across draft trees due to varying prediction difficulties in different image regions. To address this, we introduce Adjacency-Adaptive Dynamical Draft Trees, dubbed as PEANUT, which dynamically adjust draft tree depth and width by leveraging adjacent token states and prior acceptance rates. PEANUT optimizes tree construction using spatial token relationships, achieving more stable acceleration and higher acceptance rates. Evaluations on text-to-image generation show that PEANUT dramatically outperforms methods with draft tree-like EAGLE-2 in inference efficiency while preserving lossless image quality, and can also be combined with techniques such as LANTERN that relax sampling criteria.

## 1 Introduction

Autoregressive (AR) models (Sun et al., 2024; Liu et al., 2024; Tian et al., 2024) have made remarkable strides in image generation, achieving image quality that rivals or surpasses diffusion-based methods. Recent advances, such as Anole (Chern et al., 2024) and Lumina-mGPT (Liu et al., 2024), have further advanced AR models by scaling with massive multimodal data. Despite the significant potential of visual autoregressive (AR) models, a key challenge is their high computational cost during inference, stemming from the token-by-token generation process typical of AR architectures.

A typical approach for accelerating AR models is speculative decoding (Chen et al., 2023; Leviathan et al., 2023), which is an advanced inference acceleration technique designed to improve the decoding efficiency of large language models (LLMs) without compromising output quality. It operates by rapidly generating multiple draft tokens using a lightweight draft model and subsequently verifying them with the larger, more accurate target model. By speculatively precomputing several tokens and validating them in parallel batches, speculative decoding significantly reduces the number of sequential forward passes required. Recent methods like SpecInfer (Miao et al., 2024), Medusa (Cai et al., 2024), and EAGLE-2 (Li et al., 2024) adopt tree-based draft token structures, offering a larger search space than traditional linear-chain approaches. While speculative decoding has advanced LLMs, its application to visual AR models remains underexplored. To the best of our knowledge, only a few studies, such as SJD (Teng et al., 2025) and LANTERN (Jang et al., 2025), have studied speculative decoding in visual AR models. Specifically, SJD adopts a chain structure for draft tokens, generating only one draft token per position in the token sequence, which limits efficiency. LANTERN improves upon this by employing a tree structure that generates multiple draft tokens per position. However, this approach is lossy, as it relaxes speculative decoding and consequently compromises generation quality. Despite these advances, the development of a lossless and more efficient draft structure tailored for visual AR models remains an open challenge.

We observe a phenomenon during image speculative decoding generation where draft tokens tend to flock together in specific regions of the image. These uneven distributions result in significant

disparities in acceptance rates across different positions within the generation image. Consequently, as Figure 1 illustrates, it leads to inefficient utilization of the draft tree, resulting in a slowdown of the speculative decoding process in visual AR models. Specifically, we identify a key problem, namely the **imbalance building draft tree**, which significantly impedes the effective application of speculative decoding to visual AR models.

In contrast to the speculative decoding employed in existing Visual AR models, the token initialization strategy within the SJD (Teng et al., 2025) focuses on the relationships between image adjective tokens. Similarly, the concept of latent proximity permitting token interchangeability, as described in LANTERN (Jang et al., 2025), addresses the probabilistic associations among these image adjective tokens. Analogously, this paper investigates the similar associations that exist among the draft trees generated by such image adjective tokens.

To address the above issues, we propose a solution of building draft trees dubbed as PEANUT that makes use of the varying difficulty in sampling from different positions of the image to dynamically adjust the depth and top-k of the draft tree, thereby enhancing the acceptance rate and acceptance length. Specifically, we utilize the similarity between depth and probability positions of adjacent draft tokens in the draft tree to more accurately initialize the current draft tree. Then, based on the state of the previous draft trees, we adjust the expected depth and width (top-k) of draft trees through appropriate corrections. Thus, we select the depth and top-k of the draft tree more precisely to achieve a higher utilization rate of the draft tree.

Our approach achieves speed-up rate raising in the speculative decoding of the token sequence, which is equipped with the characteristics of image tokens, according to our text-conditional experiments on MSCOCO2017 (Lin et al., 2015) and parti-prompts (Yu et al., 2022).

Figure 1: The draft model faces two situations in different image regions. The image token depth matrix tracks the depth of the draft tree at which each image token resides. In this matrix, brighter areas signify deeper locations of the image tokens within the draft tree. For complex regions, the acceptance length is lower than the height of the draft tree, making unused layers wasteful and reducing the acceleration rate. A shallow draft tree is appropriate. For simple regions, the potential acceptance length exceeds the draft tree height, so building a deeper tree can increase the acceptance length and boost the acceleration rate.

To summarize, our key contributions are as follows:

- **Observation of the bottleneck in efficient visual speculative decoding**: We conduct extensive experiments and find that the imbalance in acceptance ratios across different image regions in the current draft tree constitutes the primary bottleneck in applying draft tree speculative decoding to visual AR models.

- **Novel method for dynamically building draft tree**: We design a dynamically building draft tree method, adapting the adjacent states of tokens dubbed as PEANUT. PEANUT first initializes the draft tree based on horizontally adjacent draft trees, and subsequently adjusts it according to the states of the adjacent draft trees, leading to a higher draft tree utilization rate without sacrificing image generation performance.

## 2 RELATED WORK

**Visual Autoregressive Models:** Autoregressive (AR) models have gained prominence in image generation, delivering quality rivaling diffusion models (Saharia et al., 2022) through sequential token prediction. Unlike diffusion models, visual AR models tokenize images into discrete sequences and process them with transformer architectures, the same to large language models (LLMs). Existing works like LlamaGen (Sun et al., 2024), Anole (Chern et al., 2024), and Lumina-mGPT (Liu et al., 2024) excel in text-conditional image generation, using quantized autoencoders to convert images into token sequences for transformer-based sampling.

**Speculative Decoding:** The core idea of speculative decoding (Chen et al., 2023; Leviathan et al., 2023; Chen et al., 2024) is to first draft and then verify: quickly generate a potentially correct draft

and then check which tokens in the draft can be accepted. This method first applies to large language models with AR structure. The initial draft form is the chain structure (Santilli et al., 2023; Zhao et al., 2024; Kou et al., 2024). And then SpecInfer (Miao et al., 2024) introduces a draft form with tree structure, which represents **draft tree**. The draft tree is equipped with two parameters, top-k $\hat{k}$ and depth $\hat{d}$, where $\hat{k}$ represents the number of each child node in the draft tree and $\hat{d}$ represents the depth of the draft tree. The draft form with tree structure (Miao et al., 2024; Cai et al., 2024; Li et al., 2024; Zhang et al., 2024) has flourished. From MEDUSA (Cai et al., 2024) to EAGLE-2 (Li et al., 2024), unleashing the potential of the tree structure draft tree, these methods greatly increase the speed-up ratio.

One of the few works related to speculative decoding of image token sequences is speculative decoding for Multi-LLM (Gagrani et al., 2024), which provides a simple yet efficient approach to applying speculative decoding in Multi-LLMs. With the introduction of Speculative Jacobi Decoding (Teng et al., 2025), speculative decoding has been extended to visual autoregressive (AR) models. Although the GSD (So et al., 2025) method, based on SJD, has modified its sampling paradigm, the structure of its draft token remains a chain structure. However, the draft tokens in these methods follow a chain structure rather than a tree structure. Existing draft tree methods like LANTERN (Jang et al., 2025) employ a lossy tree-structured drafting approach with relaxation of speculative decoding.

## 3 Preliminaries and Motivation

We first introduce the necessary notation. Then, we describe the motivations for PEANUT, highlighting the challenges and solutions for optimizing inference efficiency while maintaining the quality of conditional generation.

### 3.1 Notation

Drawing from LLMs, we adapt speculative decoding for image generation. An image is tokenized into a sequence $S = (s_1, s_2, \ldots, s_T)$ via a quantized autoencoder, where a lightweight encoder and quantizer produce discrete tokens $s_t \in \{1, \ldots, K\}$ (codebook size $K$), and a decoder reconstructs $\hat{I}$ from $S$. The target model $\mathcal{L}$, an autoregressive transformer, generates $S$ conditioned on a prompt $\rho$ (e.g., text or label). We define $p(s_t|s_{1:t-1}, \rho) = \mathcal{L}(s_t|s_{1:t-1}, \rho)$ as the sampling result of the conditional generation function (CFG) (Ho & Salimans, 2021) for the target model. A smaller draft model $\mathcal{R}$ generates $q(s_t|s_{1:t-1}, \rho)$ approximates the output of $\mathcal{L}$. In speculative decoding in visual AR models, given a prefix $s_{1:t-1}$ and $\rho$, $\mathcal{R}$ proposes a draft sequence $\hat{s}_{t+1:t+L}$ of length $L$, which $\mathcal{L}$ verifies in parallel. Among them, $L$ represents the total number of tokens in the draft tree. We define $\hat{s}_{ans(t)}$ as the ancestor sequence to node $\hat{s}_t$ based on the tree mask, which means $\hat{s}_{ans(t)}$ is the sequence from root to $\hat{s}_t$. The acceptance probability is:

$$r_{t+j} = \min\left(1, \frac{p\left(\hat{s}_{t+j}|s_{1:t}, \hat{s}_{ans(t+j)}, \rho\right)}{q\left(\hat{s}_{t+j}|s_{1:t}, \hat{s}_{ans(t+j)}, \rho\right)}\right), j = 1, \ldots, L \tag{1}$$

where both $p(\hat{s}_{t+j} \mid s_{1:t-1}, \hat{s}_{ans(t+j)}, \rho)$ and $q(\hat{s}_{t+j} \mid s_{1:t-1}, \hat{s}_{ans(t+j)}, \rho)$ are computed using CFG.

To further optimize drafting, we integrate a dynamic draft tree $\mathcal{T}_{\text{draft}}$, based on EAGLE-2, having depth $\hat{d}$ and width $\hat{k}$. Each node $v$ of the draft tree represents a token $s_v$ with confidence $c_v = q(s_v|s_{1:t-1}, s_{\text{anc}(v)}, \rho)$. The tree expands by selecting the top-$k_d$ nodes at depth $d$ based on path confidence $P_v = \prod_{u \in \text{Path(root}, v)} c_u$, where Path(root, $v$) is the sequence from root to $v$. For each selected node at position $(d, k)$, $\mathcal{R}$ generates $k_{d+1}$ child nodes at depth $d + 1$, positioned at $(d + 1, 1), \ldots, (d + 1, k_{d+1})$, sampling from $q(\cdot|s_{1:t-1}, s_{\text{anc}(v)}, \rho)$, with $k_{d+1} < \hat{k}$. The sequence is then reranked and verified with $\mathcal{L}$.

### 3.2 Motivation

Speculative decoding has demonstrated significant success in accelerating autoregressive (AR) models for text generation (Chen et al., 2023; Leviathan et al., 2023). Recent advancements, such as those employing draft tree structure (Miao et al., 2024; Cai et al., 2024; Li et al., 2024), have expanded the search space for draft tokens. Notably, EAGLE-2 (Li et al., 2024) introduces a dynamic candidates

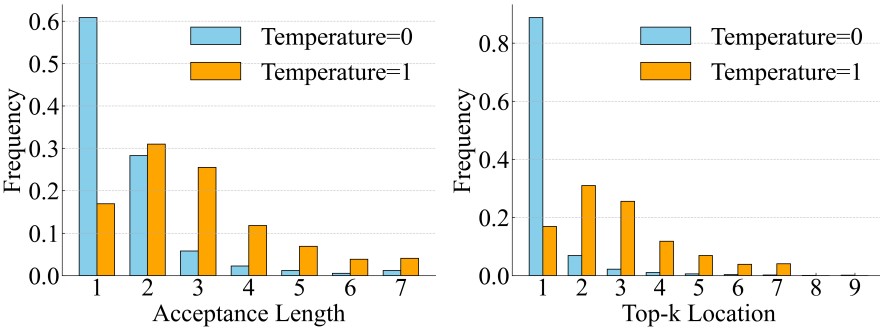

Figure 2: **(a) Left:** Frequency of acceptance lengths during speculative decoding with $\mathcal{T}_{\text{draft}}$ ($\hat{d} = 5$, $\hat{k} = 10$) over 100 image generations using Anole at $T = 0$ and $T = 1$. **(b) Right:** Frequency of the top-$k$ positions of accepted draft tokens, where 'Top-$k$ Location' denotes the minimum $k_d$ required for $\mathcal{R}$ to include the correct token in the draft phase for acceptance by $\mathcal{L}$.

token tree, $\mathcal{T}_{\text{draft}}$, with configurable depth $\hat{d}$ and width $\hat{k}$, enabling manual adjustment of the token search scale. This flexibility positions EAGLE-2 as a promising approach for accelerating visual AR models ($\mathcal{L}$), which generate token sequences $S$ conditioned on a prompt $\rho$.

However, applying EAGLE-2 to visual AR models reveals inefficiencies stemming from the expansive search scale of $\mathcal{T}_{\text{draft}}$. To investigate this, we analyze the frequency of acceptance lengths during speculative decoding with $\mathcal{L}$ and a draft model $\mathcal{R}$. Figure 2(a) illustrates the distribution of acceptance lengths over 100 image generation trials, using a draft tree configured with $\hat{d} = 7$ and $\hat{k} = 10$, under temperature settings $T = 0$ and $T = 1$. At $T = 1$, the acceptance lengths exhibit significant variance, indicating that a static $\hat{d}$ leads to inefficiencies. For instance, when the acceptance length $\tau$ is 3, constructing a tree of depth 7 wastes computational resources on four unnecessary layers. Conversely, reducing $\hat{d}$ to 3 caps $\tau$ at 3, limiting the potential acceleration in regions where $\mathcal{R}$ could predict longer sequences. This trade-off complicates the selection of an optimal $\hat{d}$ for visual AR speculative decoding, a phenomenon also noted in prior works such as SJD (Teng et al., 2025) and LANTERN (Jang et al., 2025), which highlight local similarities in token generation.

We identify a critical challenge: **imbalance in acceptance rates of draft trees**. During speculative decoding of the token sequence $S$, the acceptance length $\tau$ varies across positions due to differences in prediction difficulty for $\mathcal{R}$. This variability, depicted in Figure 2(a), suggests that a fixed-depth $\mathcal{T}_{\text{draft}}$ either overextends in regions of low $\tau$, reducing the acceptance rate $\alpha = \tau/\hat{d}$, or underextends in regions of high $\tau$, constraining the expected ratio $\mathbb{E}[\frac{\tau}{T_{\text{draft}}}]$.

Based on the above observations, we propose a potential solution: regions with simpler textures (e.g., low-frequency backgrounds) in the generated image exhibit higher $\tau$ values, as $\mathcal{R}$ can predict tokens more accurately, and when visual error tolerance is high, the distribution discrepancy between the draft model and target model is smaller. In contrast, complex texture regions (e.g., high-frequency details like fur) show lower $\tau$ values due to reduced visual error tolerance, resulting in significant distribution divergence between $q(\cdot|s_{1:t-1}, \rho)$ and $p(\cdot|s_{1:t-1}, \rho)$. This behavior is closely related to the spatial coherence of images—adjacent tokens demonstrate strong correlations in acceptance lengths, reflecting local consistency in generation difficulty. Leveraging this property, we can dynamically adjust the structure of the draft tree by analyzing the acceptance rates of neighboring regions.

Additionally, Figure 2(b) reveals variability in the top-$k$ positions of accepted tokens within $q(\cdot|s_{1:t-1}, \rho)$. In complex regions, the position of draft tokens' probabilities may rank lower in $\mathcal{R}$'s distribution compared to $\mathcal{L}$, occasionally falling outside the top-$k$ range ($k_d > \hat{k}$), leading to rejection. This discrepancy underscores the need for adaptive $\hat{k}$ alongside $\hat{d}$.

Motivated by these findings, we propose PEANUT, an algorithm that dynamically adjusts the depth $\hat{d}$ and width $\hat{k}$ of $\mathcal{T}_{\text{draft}}$ during the expansion phase of speculative decoding. By tailoring tree

Figure 3: Comparison of the process of building draft tree EAGLE-2 and PEANUT. Nodes in the same layer share the same position index. PEANUT can construct a more appropriate draft tree with the right depth and width based on the positions of the nodes in the previous layer and the status of the draft tree.

structure to the local prediction difficulty, PEANUT aims to maximize $\mathbb{E}[\frac{\tau}{T_{\text{draft}}}]$ while minimizing unnecessary computation.

## 4 PEANUT: ADJACENCY-ADAPTIVE DYNAMICAL DRAFT TREES

To address the challenge of uneven acceptance rates across draft trees at various positions, stemming from inconsistent acceptance lengths, we introduce Adjacency-Adaptive Dynamical Draft Trees, dubbed PEANUT. As shown in Figure 3, this approach dynamically builds a draft tree by adapting to the acceptance rate state of adjacent tokens in visual auto-regressive models. Let $\hat{d}$ be the depth of the draft tree and $\hat{k}$ be the width (the top-k value) of the draft tree. PEANUT constructs the draft tree through two phases: initialization and adaptation. First, the depth $\tilde{d}$ and width $\tilde{k}$ of the current draft tree are initialized according to the established strategy. Second, it revises these two values according to the acceptance rate of the previous draft trees, which reflects the current level of prediction difficulty. Details are described below.

### 4.1 ADJACENT INITIALIZATION

We introduce Adjacent Initialization, which addresses the initialization of the depth $\tilde{d}$ and top-k $\tilde{k}$ for a new draft tree associated with the image token $s^{(i,j)}$. Let $(\mathbf{i}, \mathbf{j})$ denote the position of the token to be predicted within a two-dimensional grid representing the encoded image. Three strategies that leverage adjacency to initialize $\tilde{d}$ and $\tilde{k}$ are provided as follows:

- **Horizontal Repeat (Repeat Left Adjacent Draft Tree)**: Set $\tilde{d} = d^{i,j-1}$ and $\tilde{k} = k^{i,j-1}$, using the draft tree attributes of $s^{(i,j-1)}$.
- **Vertical Repeat (Repeat Above Adjacent Draft Tree)**: Set $\tilde{d} = d^{i-1,j}$ and $\tilde{k} = k^{i-1,j}$, based on $s^{(i-1,j)}$.
- **Random Initialization**: Sample $\tilde{d} \sim \mathcal{U}(d_{\min}, d_{\max})$ and $\tilde{k} \sim \mathcal{U}(k_{\min}, k_{\max})$.

These three strategies build on experimental findings from SJD (Teng et al., 2025), which reveal that tokens positioned next to each other horizontally or vertically exhibit similar probabilities during image generation. Building on this insight, the Horizontal Repeat and Vertical Repeat strategies were developed. Meanwhile, the Random strategy was crafted for images where the correlation between adjacent tokens is comparatively weak. These simple yet effective approaches adapt $\tilde{d}$ and $\tilde{k}$ to the spatial context of $s^{(i,j)}$.

### 4.2 BISECTIONAL DYNAMIC ADAPTATION

After obtaining the initial draft tree, we design Bisectional Dynamic Adaptation to adjust the depth and width of the draft tree. As shown in Figure 4, we categorize the input draft tree into two statuses. Let $\beta$ be the positive threshold, which determines the positive or negative status.

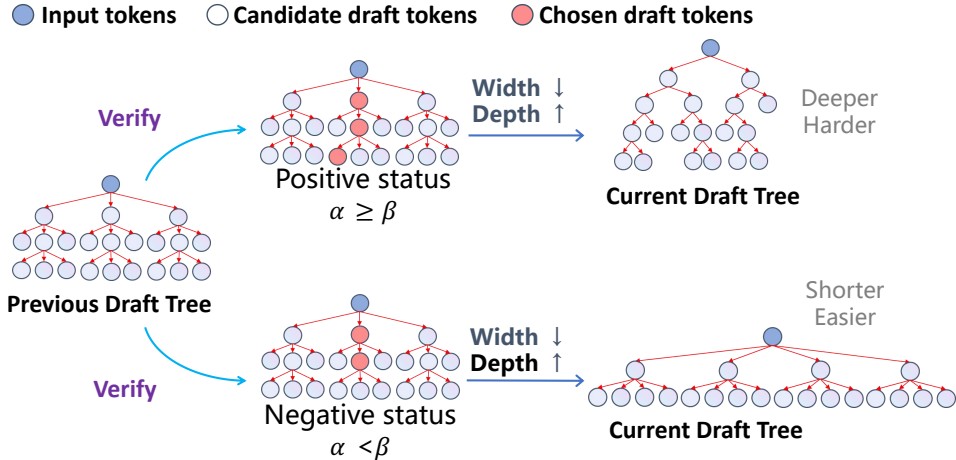

Figure 4: The adapting phase main process. According to the positive threshold $\beta$, the previous draft tree is classified as either a positive or a negative state. The width of the draft tree is decided by the setting of top-k. If it is from a complex texture to a simple texture, PEANUT's draft trees will turn deeper and narrower, which will remove unnecessary expansions. On the contrary, it will turn into a shorter and wider tree so that the current draft tree can more easily find the correct target tokens.

In the positive status, the acceptance rate $\alpha$ exceeds the threshold $\beta$, meaning that all tokens in each layer of the draft tree are fully utilized. In this scenario, the capability of the draft model is maximized from the perspective of depth, allowing it to construct deeper structures while relying less on top-k sampling.

In the negative status, the acceptance rate $\alpha$ is less than the threshold $\beta$, indicating that some tokens in certain layers are not utilized, thus limiting the effectiveness of the draft model. Consequently, the draft tree should be shallower, and a larger top-k should be used to increase the likelihood of predicting the correct token.

Therefore, the adjusted values of the depth $\hat{d}$ and the width $\hat{k}$ are computed as:

$$\hat{d} = \begin{cases} \tilde{d} + l_d, \ \alpha \geq \beta, \\ \tilde{d} - l_d, \ \alpha < \beta. \end{cases} \quad \hat{k} = \begin{cases} \tilde{k} - l_k, \ \alpha \geq \beta, \\ \tilde{k} + l_k, \ \alpha < \beta, \end{cases} \tag{2}$$

where $l_d$ and $l_k$ are the adjustment steps for the depth and width, respectively.

Using a shallower draft tree with a larger top-k expands the larger search scope for per layer. A shallower draft tree effectively drafts tokens in areas with simple textures. Conversely, when using a deeper draft tree, reducing the top-k can minimize the space and time costs associated with building the draft tree. Accurately predicting the positions of child nodes containing the accepted tokens enhances the efficiency of building draft trees, thereby increasing the generation speedup.

According to EAGLE-2 (Li et al., 2024), the time costs $C_T$ and the space peak costs $C_S$ of building dynamic draft trees are calculated as:

$$C_T = T_S \cdot \hat{d} + T_N \cdot N, \tag{3}$$

$$C_S = \hat{k}^2 \cdot (\hat{d} - 1) + \hat{k}, \tag{4}$$

where $N$ is the total number of tokens, $T_S$ denotes the time of inference of draft model, and $T_N$ denotes the time of building tree mask.

Considering the worst-case time complexity when the current token remains in the negative state and cannot return to the positive state. Meanwhile, if the negative state suddenly transitions back to the positive state, the top-k value may become excessively large. Therefore, to prevent excessive $C_T$ and $C_S$, we impose the constraints $d_{min} < \hat{d} < d_{max}$ and $k_{min} < \hat{k} < k_{max}$ to limit the depth and width of the draft tree. Additionally, we only restrict the top-k when $\tilde{d} \pm l_d > 1$, as in this case, the top-k does not increase proportionally to the square of the difference.

Table 1: The evaluation on the validation set of MSCOCO2017. Speedup ratio is denoted by $SR$, the mean acceptance length by $\tau$, the mean draft tree depth by $\bar{d}$, and the temperature by $T$.

| Method | T=0 | | | | | T=1 | | | | |
| | Acceleration | | | Image Quality | | Acceleration | | | Image Quality | |
| | SR (↑) | $\tau$ (↑) | $\bar{d}$ | HPSv2 (↑) | CLIP Score (↑) | SR (↑) | $\tau$ (↑) | $\bar{d}$ | HPSv2 (↑) | CLIP Score (↑) |
|---|---|---|---|---|---|---|---|---|---|---|
| Anole (Chern et al., 2024) | 1.00× | 1.00 | 1.00 | 0.2309 | 0.3086 | 1.00× | 1.00 | 1.00 | 0.2360 | 0.3042 |
| EAGLE-2 (Li et al., 2024) | 1.62× | 2.91 | 5.00 | 0.2338 | 0.3078 | 0.76× | 1.11 | 5.00 | 0.2361 | 0.3047 |
| LANTERN (Jang et al., 2025) | 3.03× | 4.25 | 5.00 | 0.2188 | 0.2955 | 1.38× | **2.00** | 5.00 | 0.2303 | 0.3005 |
| **PEANUT** | 2.21× | 3.40 | 3.86 | 0.2331 | 0.3081 | 1.06× | 1.10 | 2.09 | 0.2367 | 0.3047 |
| **PEANUT+LANTERN** | **3.13×** | **4.86** | 5.15 | 0.2191 | 0.2965 | **1.53×** | 1.87 | 2.10 | 0.2331 | 0.3016 |

## 5 EXPERIMENTS

### 5.1 EXPERIMENTAL SETTINGS

**Datasets:** For the text-conditional image generation, we conduct experiments on the acceleration effect on parti-prompts (Yu et al., 2022) and MS-COCO2017 (Lin et al., 2015). We utilize random 100 captions sampling from the MS-COCO2017 validation captions to evaluate the actual speedup. The same experimental setting is also conducted for Parti-Prompts.

**Evaluation Metrics:** PEANUT is a lightweight acceleration method that neither fine-tunes the target visual AR Models' weights during training nor relaxes the acceptance conditions during decoding. Thus, the generation results remain unchanged in image quality as a result of the framework of EAGLE-2 (Li et al., 2024). To measure the acceleration performance, we adopt the following metrics:

- **Speedup Ratio (SR)**: The actual test speedup ratio relative to vanilla visual auto-regressive decoding.
- **Acceptance Length ($\tau$)**: The average number of tokens generated per drafting-verification cycle, indicating the number of tokens accepted by the target visual AR Model decoding from the draft model.
- **Mean Draft Trees Depth ($\bar{d}$)**: The average depth of draft trees per drafting-verification cycle, indicating the depth of draft trees by the draft model.

**Implementation Details:** We set all generation latent size to 576 and classifier-free guidance score to 4.0. To ensure consistency and comparability with EAGLE-2, we set temperature $T \in \{0.0, 1.0\}$. To validate our method PEANUT, for Anole's draft model, we set $\beta = 1$ $l_d = 1$, and $l_k = 3$, where the depth and top-k of draft trees are limited in $(0, 10)$ and $(3, 14)$. We evaluate our approach on two different models, which are LlamaGen (Sun et al., 2024) and Anole (Chern et al., 2024). For LlamaGen's draft model, we set $\beta = 1$, $l_d = 1$ and $l_k = 10$, where the depth and top-k of draft trees are limited in $(0, 10)$ and $(3, 40)$. We evaluate each method in both the greedy decoding setting with T = 0 and the speculative decoding with T = 1. More algorithm details about greedy decoding and speculative decoding show in Appendix A, and more training details show in Appendix C.2.

**Training Implementation:** Our training implementation is based on the open source repository of EAGLE-2. To train the text-condition draft model, we randomly sample 200k text-image pairs in LAION-COCO (Kang et al., 2023) dataset for Anole's draft model, which is used to train LlamaGen-XL(stage I) (Kang et al., 2023) target model. For Anole's draft model, we directly utilize the draft models that are already available in the LANTERN (Jang et al., 2025) project. Since LlamaGen (Sun et al., 2024) uses classifier-free guidance (Ho & Salimans, 2021) to generate images, we randomly dropped 10% conditional embedding during training, consistent with target model training.

### 5.2 RESULTS OF ACCELERATED IMAGE GENERATION

Table 1 demonstrates that PEANUT achieves substantial acceleration compared to other methods in Anole. At a temperature of 0, PEANUT achieved a speedup ratio of 2.21 on MSCOCO2017, while PEANUT+LANTERN achieves a speedup ratio of 3.13. At a temperature of 1, PEANUT+LANTERN also obtain speedup ratios of 1.53. Among them, PEANUT+LANTERN

Table 2: The evaluation on the validation set of parti-prompts. Speedup ratio is denoted by $SR$, the mean acceptance length by $\tau$, the mean draft tree depth by $\bar{d}$, and the temperature by $T$.

| Method | T=0 | | | | | T=1 | | | | |
| | Acceleration | | | Image Quality | | Acceleration | | | Image Quality | |
| | SR (↑) | $\tau$ (↑) | $\bar{d}$ | HPSv2 (↑) | CLIP Score (↑) | SR (↑) | $\tau$ (↑) | $\bar{d}$ | HPSv2 (↑) | CLIP Score (↑) |
|---|---|---|---|---|---|---|---|---|---|---|
| Anole (Chern et al., 2024) | 1.00× | 1.00 | 1.00 | 0.2100 | 0.2731 | 1.00× | 1.00 | 1.00 | 0.2360 | 0.3089 |
| EAGLE-2 (Li et al., 2024) | 1.98× | 3.57 | 5.00 | 0.2113 | 0.2744 | 0.80× | 1.26 | 5.00 | 0.2360 | 0.3084 |
| LANTERN (Jang et al., 2025) | 2.82× | **4.46** | 5.00 | 0.2036 | 0.2663 | 1.90× | **2.08** | 5.00 | 0.2279 | 0.3029 |
| **PEANUT** | 2.24× | 2.79 | 3.43 | 0.2109 | 0.2741 | 1.57× | 1.17 | 2.16 | 0.2370 | 0.3104 |
| **PEANUT+LANTERN** | **3.05×** | 3.97 | 4.31 | 0.2041 | 0.2664 | **2.20×** | 1.78 | 2.70 | 0.2304 | 0.3046 |

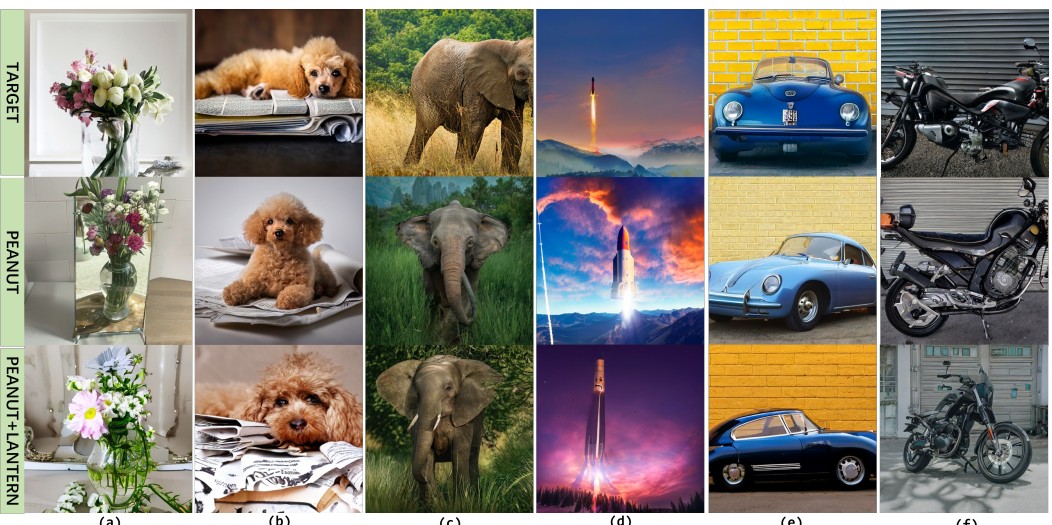

(a) "A white toilet sitting in a bathroom stall next to a TP dispenser."
(b) "A little poodle puppy laying near a newspaper with a look of guilt."
(c) "An adorable elephant walking through a grass covered forest."
(d) "A serene alpine meadow in spring:A rocket launching into space, captured at the peak of its ascent: The rocket, sleek and powerful, is shown against a backdrop of a twilight sky. Flames and smoke trail behind it as it pierces through the atmosphere. The intense light from the engines illuminates the scene, creating a breathtaking contrast with the darkening sky"
(e) "A blue Porsche 356 parked in front of a yellow brick wall."
(f) "A black Honda motorcycle parked in front of a garage."

Figure 5: Qualitative samples generated by Anole using PEANUT and standard autoregressive decoding are showcased. From top to bottom, the images correspond to outputs from standard autoregressive decoding, PEANUT (with parameters $l_d = 1$, $l_k = 3$, $\hat{d} = (0, 10)$, $\hat{k} = (3, 14)$), and PEANUT+LANTERN (with $\delta = 0.4$, $k = 1000$).

refers to PEANUT employing LANTERN's relaxed sampling for image generation. Furthermore, it can be observed that although the acceptance length $\tau$ of our method is not always the largest, its average $\bar{d}$ is smaller than that of other methods. This is precisely the result of PEANUT dynamically constructing the draft tree based on image characteristics, which saves time in building the draft tree.

Table 2 further highlights the performance of PEANUT and PEANUT+LANTERN, focusing on the efficiency of the draft tree construction. Although the acceptance length $\tau$ of our method is not always the largest, its average $\bar{d}$ is smaller than that of other methods. This efficiency stems from PEANUT's ability to dynamically construct the draft tree based on image characteristics, which reduces the time required for building the draft tree, thereby contributing to the observed acceleration in image generation.

We evaluate the generated results using various image metrics. CLIP Score (Hessel et al., 2021) and HPSv2 (Wu et al., 2023) measure the alignment quality between images and text. It can be observed that, under the same sampling methods (i.e., EAGLE-2's lossless sampling and LANTERN's relaxed sampling), PEANUT does not compromise the original sampling distribution. Figure 5 shows some images and the corresponding prompt words. In addition, we conducted measurements of other image metrics, such as FID, IS, and Aesthetic. Further details can be found in Appendix C.

## 5.3 Ablations and Analysis

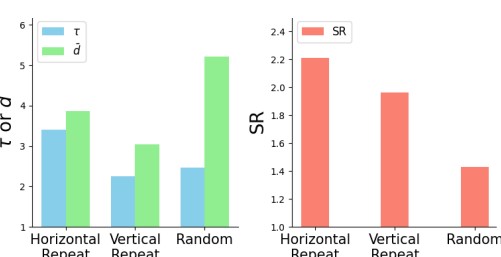

Figure 6: The influence of different initialization strategies of draft trees on the mean acceptance length $\tau$, mean draft trees depth $\bar{d}$ and speedup ratio $SR$.

Table 3: Different calculation methods of base values' calculation results on ImageNet with Temperature=1. L represents LlamaGen GPT-L, and XL stands for LlamaGen GPT-XL.

| Model | Method | SR | $\tau$ | $\bar{d}$ |
|---|---|---|---|---|
| | TokenFlock | $1.32\times$ | 2.79 | 3.60 |
| L | ($\tilde{d}$=1) | $1.27\times$ | 2.64 | 3.23 |
| 775M | ($\tilde{k}$=25) | $\mathbf{1.33\times}$ | 2.84 | 3.63 |
| | ($\tilde{d}$=1, $\tilde{k}$=25) | $1.27\times$ | 2.63 | 3.21 |
| | **PEANUT** | $1.32\times$ | **2.85** | 3.87 |
| | TokenFlock | $1.36\times$ | 2.52 | 3.35 |
| XL | ($\tilde{d}$=1) | $1.32\times$ | 2.42 | 3.12 |
| 775M | ($\tilde{k}$=25) | $1.36\times$ | **2.55** | 3.37 |
| | ($\tilde{d}$=1, $\tilde{k}$=25) | $1.32\times$ | 2.41 | 3.09 |
| | **PEANUT** | $\mathbf{1.39\times}$ | 2.53 | 3.51 |

**Impact of Initialization Strategy of Draft Trees:** We evaluate three distinct draft tree initialization strategies. As Figure 6 shows, the results demonstrate that the "Horizontal Repeat" strategy achieves a significantly higher speedup ratio. As illustrated in the left sub-figure of the accompanying Figure 6, the random strategy, with its random sampling approach, produced an excessively high $\bar{d}$, causing the acceleration effect to degrade.

Conversely, the "Vertical Repeat" strategy yields an initialization that is less precise than that of the "Horizontal Repeat" strategy. This discrepancy can be attributed to the properties of the AR model: image tokens adjacent in the horizontal direction share more similar contextual information compared to those in the vertical direction, resulting in greater similarity in their acceptance lengths. Consequently, PEANUT ultimately adopts the "Horizontal Repeat" strategy as its initialization approach.

**Impact of Calculation Methods of Initial Values:** In addition to the calculation methods of $\tilde{d}$ and $\tilde{k}$, Table 3 demonstrates several alternative approaches. Initially, we explore the incorporation of adjacent tokens in the position of the image patch following image token decoding, a method we designate as $TokenFlock$. The details of $TokenFlock$ show in Appendix C.2. Subsequently, we investigate the impact of fixing specific parameters within PEANUT, namely $\tilde{d}$ and $\tilde{k}$, to validate the rationale behind the base value calculation method. In the table, ($\tilde{d} = 1$) and ($\tilde{d} = 1$, $\tilde{k} = 25$) respectively represent the peanut method using specific fixed tree-building attributes. Our findings reveal that when either $\tilde{d}$ or $\tilde{k}$ is held constant, the acceleration ratio exhibits a certain degree of fluctuation. This fluctuation is particularly pronounced in the acceptance length, especially when $\tilde{d}$ is fixed.

## 6 Conclusion and Limitations

In this paper, we tackle the challenge of improving inference efficiency in visual autoregressive (AR) models. We identify a critical limitation of existing speculative decoding approaches when applied to visual AR models: the imbalance in draft tree acceptance rates caused by varying prediction difficulties across image regions. To address this, we propose PEANUT, an adjacency-adaptive method that dynamically adjusts draft tree depth and top-k based on the states of adjacent tokens and prior acceptance rates. Experimental results on text-conditional generation tasks demonstrate that PEANUT dramatically outperforms baselines in inference speed while maintaining generation quality. However, one limitation of our approach lies in the fact that the module of bisectional dynamic adaptation is ineffective in cases where the generated images have essentially the same acceptance lengths. In the future, we will design visual feature-oriented adaptation modules to further enhance efficiency.

## ETHICS STATEMENT

Image editing models may contain biases or occasionally produce sensitive or offensive outputs. Our models are presented strictly for academic and scientific research purposes. Any generated content does not reflect the personal views of the authors. Our work remains guided by a commitment to advancing AI technologies in ways that uphold ethical standards and resonate with societal values.

## REPRODUCIBILITY STATEMENT

For algorithms, we put the key parts in Appendix A. For datasets, we use open source datasets described in Sec. 5.1. To further ensure the reproducibility of our work, we commit to publicly releasing the complete source code and detailed experimental configuration files associated with this study upon acceptance of the paper for publication at ICLR 2026. The release will be hosted on a public code repository (e.g., GitHub) with clear documentation to guide replication of our key results.

## LARGE LANGUAGE MODELS USAGE STATEMENT

We used Large Language Models (LLMs) as auxiliary tools during the preparation of this manuscript. In particular, LLMs were employed to polish the language, improve grammar, and enhance the readability of the text. All conceptual ideas, technical contributions, analyses, and conclusions presented in this work are entirely our own and were developed independently of LLM assistance. The models were not used to generate novel scientific content, perform data analysis, or contribute to the design of experiments. We have carefully verified all statements and ensured that the final version of the manuscript accurately reflects our intended meaning and contributions.

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

APPENDIX

# A  ALGORITHM DETAILS

## A.1  GREEDY SAMPLING AND SPECULATIVE DECODING ON VISUAL AR MODEL

---
**Algorithm 1** Greedy Sampling for Token Generation
---
**Require:** Target model $\mathcal{T}$, initial context $x_{1:i-1}$, maximum length $T_{\max}$
**Ensure:** Generated sequence $x_{1:T}$
1: $t \leftarrow i$                                                    ▷ Start from the current position
2: **while** $t \leq T_{\max}$ **and** $x_t \neq$ EOS **do**
3:     $p(\cdot \mid x_{1:t-1}) \leftarrow \mathcal{T}(\cdot \mid x_{1:t-1})$                  ▷ Get token distribution
4:     $x_t \leftarrow \arg\max_x p(x \mid x_{1:t-1})$                          ▷ Greedy selection
5:     $t \leftarrow t + 1$
6: **end while**
7: **return** $x_{1:t-1}$

---

---
**Algorithm 2** Speculative decoding for Token Generation
---
**Require:** Target model $\mathcal{T}$, draft model $\mathcal{S}$, initial context $x_{1:i-1}$, draft length $L$, maximum length $T_{\max}$
**Ensure:** Generated sequence $x_{1:T}$
1: $t \leftarrow i$                                                    ▷ Start from the current position
2: **while** $t \leq T_{\max}$ **do**
3:     **Draft Phase:**
4:     Initialize draft sequence $\hat{x}_{t:t+L-1} \leftarrow []$
5:     **for** $j \leftarrow 0$ to $L - 1$ **do**
6:        Compute draft distribution: $r(\cdot|x_{1:t-1}, \hat{x}_{t:t+j-1}) = \mathcal{S}(\cdot|x_{1:t-1}, \hat{x}_{t:t+j-1})$
7:        Sample draft token: $\hat{x}_{t+j} \sim r(\cdot|x_{1:t-1}, \hat{x}_{t:t+j-1})$
8:        Append $\hat{x}_{t+j}$ to $\hat{x}_{t:t+L-1}$
9:     **end for**
10:    **Verification Phase:**
11:    Compute target probabilities: $q(\cdot|x_{1:t-1}, \hat{x}_{t:t+j-1})$ for $j = 1$ to $L$
12:    Initialize accepted sequence $a \leftarrow []$
13:    **for** $j \leftarrow 1$ to $L$ **do**
14:       Compute acceptance probability: $\alpha_j \leftarrow \min\left(1, \frac{q(\hat{x}_{t+j-1}|x_{1:t-1}, \hat{x}_{t:t+j-2})}{r(\hat{x}_{t+j-1}|x_{1:t-1}, \hat{x}_{t:t+j-2})}\right)$
15:       **if** random number $u \sim$ Uniform$(0, 1) < \alpha_j$ **then**
16:         Accept $\hat{x}_{t+j-1}$ and append to $a$
17:       **else**
18:         Compute adjusted distribution: $p_{\text{adj}}(\cdot) \propto \max(0, q(\cdot|x_{1:t-1}, a) - r(\cdot|x_{1:t-1}, a))$
19:         Sample correction: $x_{t+|a|} \sim p_{\text{adj}}(\cdot)$
20:         Append $x_{t+|a|}$ to $a$
21:         Break                                 ▷ Stop verifying further draft tokens
22:       **end if**
23:    **end for**
24:    Append $a$ to $x_{1:t-1}$
25:    $t \leftarrow t + |a|$
26:    **if** $x_t =$ EOS **then**
27:       Break
28:    **end if**
29: **end while**
30: **return** $x_{1:t-1}$

---

Greedy sampling is a straightforward method for generating sequences. At each step, it selects the token with the highest probability according to the target model and continues until a stopping condition is met. The specific detail is shown in Algorithm 1.

Speculative decoding speeds up token generation by using a smaller, faster draft model to propose multiple tokens at once. These draft tokens are then verified in parallel by the target model, reducing the number of times the target model needs to be called. The specific detail is shown in Algorithm 2.

## B   APPLICATION OF EAGLE'S DRAFT MODEL TO CLASSIFIER-FREE GUIDANCE

In this section, we outline the adaptation of EAGLE's draft model, originally designed for feature-level autoregressive prediction, to enhance image generation within the Classifier-Free Guidance (CFG) framework using LlamaGen as the target model. This approach leverages speculative decoding to accelerate inference while preserving conditional fidelity. We detail the embedding and feature representations, the draft model's prediction process, and the CFG-augmented speculative decoding mechanism, with the full workflow illustrated in Figure 7.

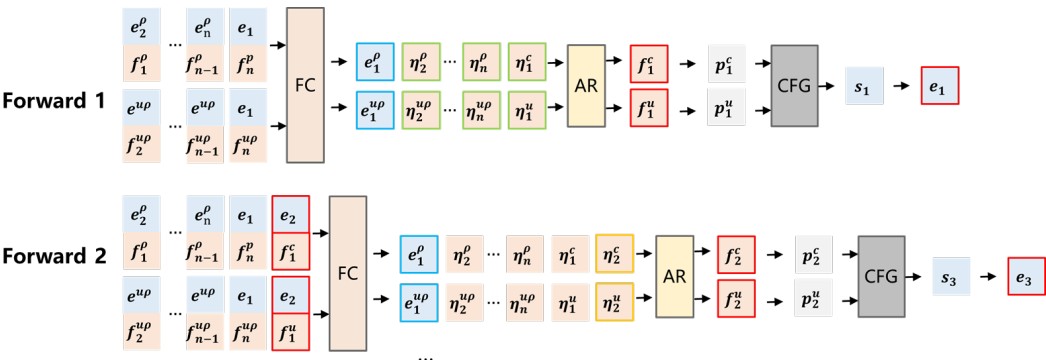

(a) Draft Model without classifier-free guidance

(b) Draft Model with classifier-free guidance

Figure 7: Illustration of classifier-free guidance draft trees and draft trees without classifier-free guidance on the framework of EAGLE. Flowchart of EAGLE's draft model application to CFG in image generation, illustrating the dual-path feature prediction, and CFG integration.

### B.1   TOKEN EMBEDDINGS AND FEATURE REPRESENTATIONS.

Consider an image tokenized into a sequence $S = (s_1, s_2, \ldots, s_T)$, where $s_t \in \{1, \ldots, K\}$ is a discrete codebook index. The target model, LlamaGen ($\mathcal{L}$), autoregressively processes this sequence, conditioned on a prompt $\rho$ (e.g., class label or text). Token embeddings are defined as:

- $e_t = \mathcal{L}_{\text{embedding}}(s_t)$, mapping $s_t$ to its embedding independently of $\rho$.

The prompt $\rho$ is embedded differently based on its type:

- **Class-Conditional Generation**: For a class label $\rho$, $e_1^\rho = \mathcal{L}_{\text{prompt}}(\rho)$ is a single learnable vector used as a prefilling token, driving the generation of $s_1, s_2, \ldots, s_T$.

- **Text-Conditional Generation**: For a text prompt $\rho$ of length $n$, $e_{1:n}^\rho = \mathcal{L}_{\text{prompt}}(\rho)$ is a sequence encoded by a pre-trained text encoder (e.g., FLAN-T5 XL) and projected (e.g., via MLP) to align with the transformer's input space, guiding $s_t$ sampling from $\mathcal{L}(s_t|s_{1:t-1}, e_{1:n}^\rho)$.

These embeddings are processed by $\mathcal{L}$'s decoder (decoder-only structure) to yield:

- Conditional feature: $f_{t-1}^c = \mathcal{L}_{\text{decoder}}(e_{1:t-1}, e_{1:n}^\rho)$,

- Unconditional feature: $f_{t-1}^u = \mathcal{L}_{\text{decoder}}(e_{1:t-1}, e_{1:n}^{u\rho})$, where $e_{1:n}^{u\rho}$ is a sequence of null embeddings matching $\rho$'s length.

The draft model $\mathcal{R}$, inspired by EAGLE-2, predicts the next feature $\hat{f}_t$, leveraging prior features and tokens to reduce uncertainty.

## B.2  DRAFT MODEL PREDICTION WITH CFG INTEGRATION.

The draft model $\mathcal{R}$ predicts features for conditional and unconditional paths to support CFG. For position $t$, intermediate tensors are computed via a fully connected (FC) layer with conditional logic:

- When $t = 1$: $\eta_1^c = \text{FC}(e_1, f_n^\rho)$,
- When $t > 1$: $\eta_t^c = \text{FC}(e_t, f_{t-1}^c)$,

where $f_n^\rho$ is the last prompt-derived feature from $e_{1:n}^\rho$ (e.g., $f_n^\rho = \mathcal{L}_{\text{decoder}}(e_{1:n}^\rho)$ at initialization). Similarly, unconditional tensors follow an analogous structure (omitted for brevity). Prompt-related intermediate features are defined recursively for $n > 1$:

- $\eta_n^\rho = \text{FC}(e_n^\rho, f_{n-1}^\rho)$,
- $\eta_n^{u\rho} = \text{FC}(e_n^{u\rho}, f_{n-1}^{u\rho})$,

where $f_{n-1}^\rho$ and $f_{n-1}^{u\rho}$ are prior prompt features (e.g., from earlier $\mathcal{L}_{\text{decoder}}$ outputs), initialized appropriately at $n = 1$. The predicted features are:

- Conditional prediction: $\hat{f}_t^c = \mathcal{R}_{\text{decoder}}(\eta_{1:t}^c, \eta_{2:n}^\rho, e_1^\rho)$,
- Unconditional prediction: $\hat{f}_t^u = \mathcal{R}_{\text{decoder}}(\eta_t^u, \eta_{2:n}^{u\rho}, e_1^{u\rho})$,

where $\eta_{1:t}^c$ is the sequence of conditional tensors up to $t$, $\eta_{2:n}^\rho$ aggregates prompt features from $e_{2:n}^\rho$, and $e_1^\rho$ anchors the initial context; $\eta_t^u, \eta_{2:n}^{u\rho}$, and $e_1^{u\rho}$ mirror this for the unconditional case. The AR model head $\mathcal{T}_{\text{AR head}}$ then yields:

- Conditional distribution:
  $q(s_{t+1}|s_{1:t}, \rho) = \text{softmax}(\mathcal{T}_{\text{AR head}}(\hat{f}_t^c))$,
- Unconditional distribution:
  $q(s_{t+1}|s_{1:t}, \emptyset) = \text{softmax}(\mathcal{T}_{\text{AR head}}(\hat{f}_t^u))$.

A draft token $\hat{s}_{t+1}$ is sampled from $q(s_{t+1}|s_{1:t}, \rho)$, forming the speculative sequence $\hat{s}_{t:t+m-1}$.

## B.3  IMAGE TOKENIZATION: ENCODING AND DECODING

The process of encoding an image into tokens and decoding tokens back into an image is central to the auto-regressive framework. This is achieved using a quantized autoencoder architecture consisting of an encoder, a quantizer, and a decoder.

1. **Encoding**: The encoder $E : \mathbb{R}^{H \times W \times 3} \to \mathbb{R}^{h \times w \times D}$ maps the image $x$ to a feature map $f = E(x)$, where $D$ is the feature dimension. The quantizer then maps each feature vector

$f^{(i,j)} \in \mathbb{R}^D$ to the nearest codebook vector $z^{(i,j)} \in Z$, with the index denoted as $t^{(i,j)}$. Formally,

$$t^{(i,j)} = \arg \min_{k \in \{1,\dots,K\}} \|f^{(i,j)} - z_k\|_2^2,$$

where $z_k$ is the $k$-th vector in the codebook $Z$, and $\|\cdot\|_2$ denotes the Euclidean norm.

2. **Decoding**: The decoder $D : \mathbb{R}^{h \times w \times C} \to \mathbb{R}^{H \times W \times 3}$ reconstructs the image $\hat{x}$ from the quantized feature map $z$, where $z^{(i,j)} = z_{t^{(i,j)}}$ is retrieved from the codebook using the index $t^{(i,j)}$. The reconstructed image is given by:

$$\hat{x} = D(z).$$

## C  MORE EXPERIMENTS AND SETTING

### C.1  MORE EXPERIMENTS

Table 4: The evaluation on the validation set of MSCOCO2017. Speedup ratio is denoted by $SR$, the mean acceptance length by $\tau$, the mean draft tree depth by $\bar{d}$, and the temperature by 1.0.

| Method | Acceleration | | | Image Quality | | | | |
|---|---|---|---|---|---|---|---|---|
| | SR (↑) | $\tau$ (↑) | $\bar{d}$ | CLIP Score (↓) | HPSv2 (↑) | IS (↑) | Aesthetic (↑) | FID (↓) |
| Anole (Chern et al., 2024) | 1.00× | 1.00 | 1.00 | 0.3042 | 0.2360 | 30.25 | 5.93282 | 20.52 |
| EAGLE-2 (Li et al., 2024) | 0.76× | 1.11 | 5.00 | 0.3047 | 0.2361 | 29.87 | 5.93804 | 20.45 |
| LANTERN (Jang et al., 2025) | 1.38× | **2.00** | 5.00 | 0.3005 | 0.2303 | 27.30 | 5.81699 | 23.65 |
| **PEANUT** | 1.06× | 1.10 | 2.09 | 0.3047 | 0.2367 | 29.54 | 5.92874 | 22.20 |
| **PEANUT+LANTERN** | **1.53×** | 1.87 | 2.10 | 0.3016 | 0.2331 | 28.25 | 5.86109 | 22.36 |

Table 5: The evaluation on the validation set of parti-prompts. Speedup ratio is denoted by $SR$, the mean acceptance length by $\tau$, the mean draft tree depth by $\bar{d}$, and the temperature by 1.0.

| Method | Acceleration | | | Image Quality | | | |
|---|---|---|---|---|---|---|---|
| | SR (↑) | $\tau$ (↑) | $\bar{d}$ | CLIP Score (↓) | HPSv2 (↑) | IS (↑) | Aesthetic (↑) |
| Anole (Chern et al., 2024) | 1.00× | 1.00 | 1.00 | 0.3089 | 0.2360 | 22.44 | 5.77940 |
| EAGLE-2 (Li et al., 2024) | 0.80× | 1.26 | 5.00 | 0.3084 | 0.2360 | 21.66 | 5.78878 |
| LANTERN (Jang et al., 2025) | 1.90× | **2.08** | 5.00 | 0.3029 | 0.2279 | 19.49 | 5.65854 |
| **PEANUT** | 1.57× | 1.17 | 2.16 | 0.3104 | 0.2370 | 22.34 | 5.80798 |
| **PEANUT+LANTERN** | **2.20×** | 1.78 | 2.70 | 0.3046 | 0.2304 | 19.94 | 5.71881 |

Table 4 illustrates that PEANUT achieves significant acceleration compared to other methods in Anole for image generation. Specifically, it presents a comparison of different methods, highlighting PEANUT's acceleration ratio of 1.06 on the MSCOCO dataset. When integrated with LANTERN's relaxed sampling, PEANUT further improves, achieving an acceleration ratio of 1.53.

Table 5 similarly demonstrates PEANUT's superior performance on the Parti-Prompts dataset. The comparison of acceleration and quality results shows that PEANUT attains an acceleration ratio of 1.57. With the incorporation of LANTERN's relaxed sampling, this ratio increases to 2.20, underscoring PEANUT's enhanced efficiency in image generation.

We evaluate the generated results using various image metrics. CLIP Score and HPSv2 measure the alignment quality between images and text. IS (Inception Score) is a metric for assessing the diversity and quality of generated images, Aesthetic evaluates the aesthetic quality of images, while FID (Fréchet Inception Distance) (Heusel et al., 2017) measures the similarity between generated and real images. Since the Parti-Prompts dataset lacks real images, Table 5 does not provide FID results. It can be observed that, under the same sampling methods (i.e., EAGLE-2's lossless sampling and LANTERN's relaxed sampling), PEANUT does not compromise the original sampling distribution.

### C.2  TOKENFLOCK

It utilizes a distinct search range defined as follows: The position of a past image token $s^{(x_j, y_j)}$ is denoted by $(\mathbf{x_j}, \mathbf{y_j})$, and the current predicted position is $(\mathbf{x_i}, \mathbf{y_i})$. We can derive a set $\Omega$ containing

positions that fall within the $\delta$-range of the current token as $\Omega = \{t_j \mid \sqrt{(x_i - x_j)^2 + (y_i - y_j)^2} \leq \delta, j < i\}$ where the hyper-parameter $\delta$ is used to select the top $\delta$ most adjacent tokens. We calculate the initial depth $\tilde{d}$ by $\tilde{d} = \sum_{t_j \in \Omega} d_j \cdot norm\left(\frac{\delta - (y_i - y_j) + 1}{\delta}\right)$ and initial $\tilde{k}$ by $\tilde{k} = \sum_{t_j \in \Omega} k_j \cdot norm\left(\frac{\delta - (y_i - y_j) + 1}{\delta}\right)$ where $norm(\cdot)$ represents a normalization function.

