# OpenReview forum: "Fast Inference of Visual Autoregressive Model with Adjacency-Adaptive Dynamical Draft Trees"
_ICLR.cc/2026/Conference — ICLR 2026 Conference Withdrawn Submission_

### Official Review · Reviewer_oEGR · 2025-10-28

**Soundness:** 3
**Presentation:** 3
**Contribution:** 2
**Rating:** 4
**Confidence:** 4

**Summary:**

This paper is about method for accelerating inference speed of autoregressive image generation, by improving the EAGLE-2. Specifically, the paper argues that the acceptance difficulty of speculative decoding significantly varied across different patch regions, and thus fixed-shape token tree cannot achieve optimal performance. To solve this problem, PEANUT introduces two novel techniques, Adjacent initialization, which initialize a token tree shape based on neighboring position's tree shape, and Bisectional Dynamic adaptation, which adjust the tree's shape based on the preceding acceptance rate. Experimental results show performance improvement compared to EAGLE-2.

**Strengths:**

- The paper is well written and easy to understand.
- The method is simple and straightforward to implement.
- It shows performance improvements over both EAGLE-2 and recent EAGLE-2 based methods(LANTERN)

**Weaknesses:**

- **Low performance** : PEANUT generally only shows acceleration at T=0 ( greedy decoding). However, as shown in Tab. 1,2 , T=0 sampling incurs a significant generation quality drops, and thus is not a standard sampling method in many AR image generation models.
At normal T=1 sampling, while quality is maintained, PEANUT only shows a low speed-up of ~1.05x.

- **Comparison with SJD** : A further issue is that SJD already achieves high acceptance rate (>2) at T=1, even without complex tree attention or separate draft model. There is no clear motivation to adopt this tree-based Speculative decoding for Image generation.

- **Novelty** : The methods for adjusting tree shape rely too heavily on heuristics, lacking clear theoretical motivation.

**Questions:**

- Can you conduct comparison with SJD at various temperatures (ie T=1), and explain the advantages of PEANUT over SJD ?
- **Related works** : Does prior research exist on dynamic tree shape adjustment for EAGLE, or is this study the first work ?

---

### Official Review · Reviewer_uzQg · 2025-11-01

**Soundness:** 3
**Presentation:** 1
**Contribution:** 1
**Rating:** 2
**Confidence:** 4

**Summary:**

This paper proposes an acceleration framework for visual AR models inspired by speculative decoding in LLMs. Traditional speculative decoding employs fixed-depth tree or chain structures, but PEANUT introduces Adjacency-Adaptive Dynamical Draft Trees, which dynamically adjust both tree depth and width based on local acceptance rates and spatial token adjacency. The method preserves lossless generation quality while achieving speedups on LlamaGen and Anole.

**Strengths:**

The paper effectively demonstrates the advantages of an adaptive speculative decoding tree within the autoregressive (AR) image generation domain, highlighting how dynamic depth and width control can improve token efficiency.

The authors’ explicit commitment to open-sourcing the code upon acceptance is commendable, as it will likely foster reproducibility and stimulate further research in AR acceleration.

**Weaknesses:**

The core idea of dynamically adjusting speculative decoding parameters is already explored in LLM works such as Cascade Drafting (Chen et al., 2024) and Medusa (Cai et al., 2024). The paper should better clarify how PEANUT goes beyond simply transferring these ideas to the visual domain.

The experimental section lacks comparisons with the most recent AR acceleration methods (e.g., parallel speculative decoding, hybrid prefix caching), making it difficult to judge the relative performance gains.

Several hyperparameters (e.g., β, ld, lk) appear empirically chosen without sensitivity analysis. The paper would benefit from a systematic study showing performance stability across different configurations.

The paper’s writing is notably unpolished, for instance, definitions of AR models are repeated unnecessarily, transitions between sections are unclear, and key concepts are buried in dense notation. Substantial editing is needed for readability. Figure 4 appears incorrectly drawn or mislabeled, which undermines understanding of the proposed architecture. Clearer visual explanations of the dynamic tree mechanism are essential.

**Questions:**

How does PEANUT fundamentally differ from adaptive speculative decoding in LLMs (e.g., Cascade Drafting or Medusa)? What specific challenges or inductive biases of the visual autoregressive domain required new algorithmic design rather than simple adaptation?

Can the authors provide empirical evidence showing that adjacency-aware adaptation (vs. random or confidence-only adaptation) improves acceptance rate or speedup? A visualization of spatial token acceptance distributions before and after applying PEANUT would be convincing.

Have the authors tested PEANUT on larger or structurally different visual AR models (e.g., Lumina-mGPT, MaskGIT variants)? This would show whether the method scales or depends heavily on specific architectures.

---

### Official Review · Reviewer_mQVD · 2025-11-01

**Soundness:** 3
**Presentation:** 2
**Contribution:** 3
**Rating:** 4
**Confidence:** 4

**Summary:**

The manuscript proposes PEANUT (Adjacency-Adaptive Dynamical Draft Trees), which is a speculative decoding method for visual autoregressive (VAR) image generation that tailors draft tree depth and width on the fly. Motivated by the observation that acceptance depth varies across image regions (shallower in complex areas, deeper in simple ones). PEANUT leverages adjacent token’s prior acceptance behavior and bisectional adaptation to decide when to deepen or widen the tree. Across benchmarks and settings, it delivers strong performance, reporting ~2.2–3.1× speedups (larger with lossy decoding) while keeping image quality roughly unchanged relative to lossless baselines, outperforming existing speculative decoding methods.

**Strengths:**

- The problem statement and the solution are inuitive and clear.
- PEANUT effectively accelerates generation and can be combined with other acceleration methods.

**Weaknesses:**

- The experimental setup does not specify the inference engine or system configuration. This omission makes the reported gains difficult to interpret. Also, reporting the absolute wall-clock latency would be informative.
- The evaluation suite omits more advanced benchmarks such as GenEval. Including such metrics would better substantiate quality under acceleration.
- In the experiments section, LlamaGen is mentioned, but LlamaGen results are absent from the main tables and discussion. Presenting LlamaGen metrics or other related baselines would increase the credibility of the manuscript.

**Questions:**

Please refer to the weaknesse section.

**Details Of Ethics Concerns:**

For another previous conference venue, I have reviewed a manuscript, which I believe was a previous version of this manuscript submission (the title of the manuscript is the same). In that previous venue's submission, the **previous** version of this manuscript was desk-rejected due to the discovery of a **prompt-injection** attack. Since the previous version was desk-rejected in the previous venue, I think that the **present** version of the manuscript is highly unlikely to contain such an attack(due to author revision), but I believe an inspection against prompt-injection attack should be conducted for this manuscript, just to be safe.

---

### Official Review · Reviewer_mV8k · 2025-11-05

**Soundness:** 2
**Presentation:** 2
**Contribution:** 1
**Rating:** 2
**Confidence:** 4

**Summary:**

This paper proposes PEANUT, a methodology to accelerate speculative decoding on visual autoregressive models. It is based on the observation that token generation difficulty varies, allegedly depending on the high- and low-frequency regions of an image, and that this difficulty exhibits locality. Based on this, PEANUT dynamically adjusts the width and depth of the draft tree using the actual acceptance rates of nearby tokens as feedback. This adaptive strategy aims to improve overall generation speed by reducing the redundant computational load.

**Strengths:**

1. The paper is well-written, and the proposed idea is easy to follow.
2. The problem of accelerating visual autoregressive model inference is timely and relevant.

**Weaknesses:**

1. **Justification for Core Observations**: The paper's motivation rests on two observations: (1) token generation difficulty varies (allegedly easier for low-frequency regions and harder for high-frequency regions), and (2) this difficulty exhibits locality. However, both the novelty and the evidence for these claims could be further substantiated.
	* The observation that token difficulty varies seems to echo existing concepts. The dynamic tree drafting in EAGLE-2, for instance, is already designed to handle this by adjusting the tree structure (width and depth) based on the drafter's confidence, which serves as a proxy for difficulty. This work appears to re-contextualize this known phenomenon using "frequency" terminology.
	* The empirical support could be strengthened. Figure 1 provides anecdotal evidence from a single example, which makes it difficult to generalize this observation. Figure 2 demonstrates that acceptance lengths and top-k positions vary, a known phenomenon in this field. It is not immediately clear how this variance establishes a causal link to the paper's specific claims about high/low-frequency regions or locality. The observations, therefore, come across as potential post-hoc justifications rather than foundational discoveries.

2. **Novelty of the Proposed Method**: The proposed method, PEANUT, appears to be a highly incremental variant of the existing EAGLE-2 framework, which raises some questions about the extent of its contribution.
	* The core mechanism (dynamically adjusting tree width and depth based on the acceptance rates of nearby tokens) appears to be a simple, local heuristic. This seems to be more of a minor tuning on top of EAGLE-2's existing dynamic tree construction, rather than a new approach to tree construction.
	* Other components, such as the Horizontal and Vertical Repeat techniques, are noted to have been proposed in prior work (e.g., SJD), which may further diminish the method's perceived originality.

3. **Concerns Regarding Experimental Evaluation**: The experimental setup could be improved to provide a more compelling and clear validation of the method's effectiveness.
	* The method was evaluated on only a single model (Anole). This makes it difficult to assess whether the findings can be generalized to other model architectures or scales.
	* A key concern is the fairness of the comparison. The paper explicitly states that PEANUT's speedup comes from reducing redundant computations (i.e., a lower average draft size, $\bar d$), not from boosting the acceptance rate. A more convincing comparison would involve benchmarking against baselines (EAGLE-2, LANTERN) configured to have a similarly low computational budget ($\bar d$). For example, a static, shallow EAGLE-2 (e.g., d=3) might achieve a similar $\bar d$ and performance. The current experiments compare PEANUT to baselines operating at a much higher, less efficient $\bar d$, which may inadvertently favor PEANUT.
	* Furthermore, even under these potentially favorable experimental conditions (where baselines perform far more computation), the reported performance gains from PEANUT appear marginal. This makes it difficult to ascertain whether the proposed adaptive strategy is truly effective or if its overall benefit is limited.

**Questions:**

See weaknesses.

---

### Note · Authors · 2025-11-19

I have read and agree with the venue's withdrawal policy on behalf of myself and my co-authors.